# Cefiderocol for Carbapenem-Resistant Bacteria: Handle with Care! A Review of the Real-World Evidence

**DOI:** 10.3390/antibiotics11070904

**Published:** 2022-07-06

**Authors:** Pasquale Sansone, Luca Gregorio Giaccari, Francesco Coppolino, Caterina Aurilio, Alfonso Barbarisi, Maria Beatrice Passavanti, Vincenzo Pota, Maria Caterina Pace

**Affiliations:** 1Department of Woman, Child and General and Specialized Surgery, University of Campania “Luigi Vanvitelli”, 80138 Naples, Italy; lucagregorio.giaccari@gmail.com (L.G.G.); francesco.coppolino@unicampania.it (F.C.); caterina.aurilio@unicampania.it (C.A.); mariabeatrice.passavanti@unicampania.it (M.B.P.); vincenzo.pota@unicampania.it (V.P.); mariacaterina.pace@unicampania.it (M.C.P.); 2Department of Translational Medical Science, Telematic University Pegaso, 80138 Naples, Italy; alfonso.barbarisi@unicampania.it

**Keywords:** healthcare-associated infections, carbapenem-resistant organisms, carbapenem-resistant enterobacteriaceae, cefiderocol

## Abstract

(1) Background: healthcare-associated infections are one of the most frequent adverse events in healthcare delivery worldwide. Several antibiotic resistance mechanisms have been developed, including those to carbapenemase. Cefiderocol (CFD) is a novel siderophore cephalosporin designed to treat carbapenem-resistant bacteria. (2) Methods: we performed a systematic review of all cases reported in the literature to outline the existing evidence. We evaluated real-world evidence studies of CFD in the treatment of carbapenem-resistant (CR) bacteria. (3) Results: a total of 19 publications treating cases of infection by CR bacteria were included. The three most frequent CR pathogens were *Acinetobacter baumannii*, *Pseudomonas aeruginosa*, and *Klebsiella pneumoniae*. A regimen of 2 g every 8 h was most frequently adopted for CFD with a mean treatment duration of 25.6 days. CFD was generally well tolerated, with fewer side effects. The success rate of CFD therapy was satisfactory and almost 70% of patients showed clinical recovery; of these, nearly half showed negative blood cultures and infection-free status. (4) Conclusions: This review indicates that CFD is active against important GN organisms including *Enterobacteriaceae, P. aeruginosa,* and *A. baumannii*. CFD seems to have a safe profile.

## 1. Introduction

According to the World Health Organization (WHO), healthcare-associated infections (HAI) are one of the most frequent adverse events in healthcare delivery worldwide [1]. Around the world, hundreds of millions of patients are affected by HAI every year, leading to a significant impact on morbidity, mortality and quality of life and representing an economic burden on healthcare systems [1].

Antimicrobial resistance (AMR) is one of the greatest global public health challenges of our time [2]. The emergence and spread of pathogens that have acquired new drug resistance mechanisms, leading to antimicrobial resistance, are a threat to our ability to treat common infections [2].

Since the spread of Gram-negative (GN) bacteria that produce extended-spectrum ß-lactamase (ESBL) enzymes, which, in addition to penicillins, confer resistance to cephalosporins and monobactam [3], carbapenems have been used as the rescue therapy to treat this type of infection [4]. Carbapenems belong to the class of β-lactam antibiotic agents which are very effective against severe and high-risk bacterial infections, and is normally a Drug of Last Resort (DoLR) antibiotic for Multi-Drug-Resistant (MDR) bacterial infections. They bind to the penicillin-binding proteins, preventing the synthesis of the bacterial cell wall.

Unfortunately, shortly after the introduction of carbapenems, GN-bacteria rapidly developed resistance to these antibiotics and spread around the world [5]. These include the Carbapenemase-Producing Organisms (CPOs) and among them the Carbapenem-Resistant Enterobacteriaceae (CREs) [6]. CREs are a group of GN bacteria, from the family of *Enterobacteriaceae*, which have developed resistance to the carbapenem antibiotics. Other CPOs include some opportunistic bacteria that have the ability to produce carbapenemase enzymes, such as *Pseudomonas aeruginosa* and *Acinetobacter baumannii.*

Several carbapenem resistance mechanisms have been developed by GN-bacteria, including the production of carbapenemase enzymes, which is nowadays identified worldwide [7]. Carbapenemase enzymes that CPOs produce include: (1) *K. pneumoniae carbapenemase* (KPC); (2) *New Delhi Metallo-beta-lactamase* (NDM); (3) *Verona integron-encoded metallo-β lactamase* (VIM); (4) *imipenemase metallo-β-lactamase* (IMP); (5) *oxacillinase* (OXA) including OXA-23, OXA-24/40, OXA-48, OXA-51, OXA-58, and OXA-143 subgroups [8].

In 2017, 8.3% of the patients who stayed in intensive-care units (ICUs) in Europe presented at least one acquired HAI (pneumonia, bloodstream infection, or urinary tract infection) [9]. Carbapenem resistance was reported in 15% of *Klebsiella* spp. isolates, 26% of *P. aeruginosa* isolates and 64% of *Acinetobacter baumannii* isolates [9]. These are a matter of national and international concern as they are an emerging cause of HAI that pose a significant threat to public health [10].

In 2017, the WHO developed a global priority pathogens list of antibiotic-resistant bacteria to help in prioritizing the research and development of new and effective antibiotic treatments. Carbapenem-resistant *Enterobacteriaceae*, carbapenem-resistant *Pseudomonas aeruginosa*, and carbapenem-resistant *Acinetobacter baumannii* are in the highest priority category [11].

It is important to identify patients with risk factors for developing MDR infections, to ensure early molecular or microbiological diagnoses and faster and more appropriate treatment [12]. The need for new antibiotics in carbapenem-resistant infections has been recognized globally. There are various antibiotics whose activity has been tested against carbapenemase-resistant microorganisms [13,14]. Cefiderocol (CFD), a novel siderophore cephalosporin designed to treat carbapenem-resistant bacteria, has shown potent in vitro activity against CPOs, including CREs [15,16,17,18,19].

In 2019, the U.S. Food and Drug Administration (FDA) granted CFD authorization to treat complicated urinary tract infections (cUTI) with limited or no alternative treatment options [20]. As of 2020, CFD is recommended for the treatment of hospital-acquired bacterial pneumonia (HABP) and ventilator-associated bacterial pneumonia (VABP), caused by GN microorganisms [20].

**Objectives**. We performed a systematic review of the cases reported in the literature to outline the existing evidence. To our knowledge, there are no previously published systematic reviews that have evaluated real-world evidence studies of CFD in the treatment of carbapenem-resistant bacteria. A systematic review of case reports cannot support the efficacy of using CFD for the treatment of carbapenem-resistant bacterial infections, but it may identify rare or unrecognized associations and may generate hypotheses for subsequent studies.

## 2. Materials and Methods

### 2.1. Protocol and Registration

Our systematic review is based on the PRISMA (Preferred Reporting Items for Systematic Reviews and Meta-Analyses) guidelines [21]. The protocol was not published, but is available if requested. The review was not registered with the international prospective register of systematic reviews (PROSPERO).

### 2.2. Literature Search Strategy

Medline, EMBASE, PubMed, Google Scholar, and The Cochrane Library-CENTRAL were screened to identify case reports of patients with carbapenem-resistant bacteria infections treated with CFD. Other studies were identified from the reference lists. We used a combination of terms such as “*cefiderocol*”, “*carbapenem resistant*”, “*Enterobacteriaceae*”, “*Pseudomonas aeruginosa*” and “*Acinetobacter baumannii*”. The titles and abstracts were screened by two researchers (P.S. and L.G.G.) to identify the keywords. The selected papers were read in full by the two independent reviewers, and if they disagreed a third reviewer (M.C.P.) was consulted.

The initial search was performed on 1 February 2022. All publications were included since inception up until the end of January 2022.

All the papers with available full text, reporting original data of patients with Carbapenem-resistant bacteria infections treated with CFD, of any age, gender, and in any setting, were included. No language restrictions were applied.

### 2.3. Inclusion and Exclusion Criteria

Studies were included if they met all of the following criteria:-The full study was published;-The study described clinical use of CFD for HAI;-The agent responsible for the infection was carbapenem-resistant bacteria;-The study reported the clinical outcome of the patient(s) treated with CFD.

Exclusion criteria were:The study did not report clinical outcome;The study had duplicate data with others (in these cases, only the largest study was retained);The study presented pooled data that did not allow for extrapolation of useful information.

According to an international expert proposal, the definition of Carbapenem-Resistant Organisms (CRO) is as follows [22,23]:Resistant to any carbapenem antimicrobial (i.e., minimum inhibitory concentrations [MIC] of ≥4 mcg/mL for doripenem, meropenem, or imipenem OR ≥ 2 mcg/mL for ertapenem);Documented to produce carbapenemase (e.g., KPC, NDM, VIM, IMP, OXA-48).

In addition, for bacteria that have intrinsic imipenem nonsusceptibility (i.e., *Morganella morganii*, *Proteus* spp., *Providencia* spp.), resistance to carbapenems other than imipenem is required.

In Table 1, we reported the breakpoint values of minimum inhibitory concentration for carbapenems according to guidelines in Europe (EUCAST) and the United States (CLSI).

## 3. Results

Based on the inclusion criteria, a total of 19 publications treating cases of infection by CR bacteria in patients were selected to conduct this review. Eight series (a randomized controlled trial [24], two retrospective monocentric [25,26], and five case series [26,27,28,29,30]) and eleven case reports published between 2017 and 2021 were included. As shown in Figure 1, the flow diagram reports the results from the literature search and the study selection process.

***Study Characteristics***. In Table 2, all the studies are presented in alphabetical order with a brief clinical description for each case.

**Patients**. A total of 172 patients were included in this review. Bassetti et al., reported 101 patients with serious infections caused by carbapenem-resistant Gram-negative bacteria, but of these only 80 completed CFD therapy. The mean age was 58.8 ± 15.8 years (15–92 years). Sex was reported for 160 patients: males were 111 (69.4%) and females were 49 (30.6%).

The most common comorbidities were: renal disease (*n* = 44, 3 of whom underwent a renal transplant), diabetes (*n* = 44), chronic pulmonary disease (*n* = 43), cancer (*n* = 31), congestive/ischemic heart disease (*n* = 17), arterial hypertension (*n* = 16), vascular disease (*n* = 13), hepatitis (*n* = 12), cerebral or neurological disease (*n* = 6), blood disease (*n* = 3), atrial fibrillation (*n* = 2), hypothyroidism (*n* = 1), pancreatitis (*n* = 1), endocarditis (*n* = 1), gout arthritis (*n* = 1), and recurrent infections in hip replacement (*n* = 1).

Primary infections were contracted in 12 different countries, including Afghanistan (*n* = 1, [32]), Australia (*n* = 1, [25]), Belgium (*n* = 1, [36]), Columbia (*n* = 1, [39]), France (*n* = 2, [28,31]), Italy (*n* = 5, [23,26,28,29,35,37]), Kuwait (*n* = 1, [34]), Nigeria (*n* = 1, [21]), Serbia (*n* = 1, [39]), Spain (*n* = 1, [24]), Thailand (*n* = 1, [39]), and US (*n* = 4, [27,33,38]). In Bassetti et al., patients came from 16 different countries in North America (*n* = 6), South America (*n* = 9), Europe (*n* = 57) and Asia (*n* = 29).

Baseline characteristics of patients are reported in Table 3.

**Type of infection and isolate**. As shown in Figure 2, CFD was used to treat respiratory infections (*n* = 84), bacteremia or sepsis (*n* = 40), osteo-articular infections (*n* = 12), complicated urinary infection (*n* = 28), intra-abdominal infections (*n* = 8), skin infection (*n* = 2), meningitis (*n* = 1) and endocarditis (*n* = 1). Falcone et al., Meschiari et al., and Rando et al., reported 22 patients with COVID-19 infection and among them 20 were admitted to ICU due to COVID-19 infection complications.

The main bacterial agents involved were: *Acinetobacter baumannii* (*n* = 70), *Pseudomonas aeruginosa* (*n* = 47), *Klebsiella Pneumoniae* (*n* = 36), *Stenotrophomonas maltophilia* (*n* = 6), *Enterobacter cloacae* (*n* = 3), *Escherichia Coli* (*n* = 3), *Acinetobacter nosocomialis* (*n* = 2), *Acinetobacter xylosidans* (*n* = 1), and *Enterobacter hormaechei* (*n* = 1).

Carbapenemase enzymes production was the only mechanism described with the predominance of OXA (oxacillin-hydrolyzing) enzyme (*n* = 15) followed by NDM (New Delhi metallo-ß-lactamases (MBL)) (*n* = 7), VIM (Verona integron-encoded MBL) (*n* = 6), and KPC (K. Pneumoniae carbapenemase) (*n* = 2) enzymes.

Significant risk factors frequently associated with carbapenem resistance were: longer hospital stays and ICU hospitalization (*n* = 80), at least three types of antimicrobial therapy received (*n* = 106), previous use of carbapenem (*n* = 70), septic shock or immunocompromised conditions (*n* = 57), and invasive life support (*n* = 11).

Susceptibility patterns (minimal inhibitory concentration (MIC) or zone of inhibition (ZOI)) of treated Gram-negative bacteria are reported in Table 4. Cefiderocol MIC values were 0.51 μg/mL for carbapenem-resistant *Acinetobacter baumannii*, 1.7 μg/mL for carbapenem-resistant *Klebsiella pneumoniae*, and 1.63 μg/mL for carbapenem-resistant *Pseudomonas aeruginosa.*

Six patients showed *Pseudomonas aeruginosa* resistant to CFD [31,36]. In Bleibtreu et al., five isolates were classified as non-susceptible to CFD (four categorized as resistant and one as intermediate). Grande Perez et al. reported resistance to CFD after 128 days of initial isolation.

**Therapeutic regimen**. A regimen of 2 g every 8 h was used for 40 patients [25,27,28,29,31,32,36,38,39,40]. Basetti et al. used a regimen of 2 g every 8 h, with dosage adjustments for altered renal function. Other therapeutic regimens were adapted according to renal function. Three studies [30,34,42] did not report CFD regimen. The mean treatment duration was 26.6 ± 23.7 days (4–102 days). Bleibtreu et al. and Edgeworth JD et al., did not report CFD administration duration.

62.3% (91/146) of patients received monotherapy; 37.7% (55/146) received combination therapy. In total, 21 patients (38.2%) of 55 received colistin-based treatment; other antibiotics administered were fosfomycin (*n* = 12), ceftazidime/avibactam (*n* = 6), tigecycline (*n* = 4), metronidazole (*n* = 3), amikacin (*n* = 2), ciprofloxacin (*n* = 2), amphotericin B, aztreonam, levofloxacin, linezolid, moxifloxacin, rifampicin, teicoplanin and tobramycin (*n* = 1).

**Adverse Events (AEs) and outcome**. As shown in Figure 3, a total of 98 treatment-emergent adverse events (TEAE) were reported. The most frequently reported TEAEs were diarrhea (*n* = 19), pyrexia (*n* = 14), septic shock (*n* = 13), and vomiting (*n* = 13). In Bassetti et al., TEAEs led to study drug discontinuation in three patients due to pyrexia, aminotransferase increase, and skin rash. In Bodro et al., cefiderocol was discontinued due to thrombocytopenia.

Finally, 100 patients (66.7%) were cured with CFD; 42 patients (28%) did not respond to therapy. A sub-analysis of outcome by type of primary infection showed that 47 patients (61%) with HAP/VAP reported clinical recovery and 30 (30%) clinical failure; among patients with other primary infections, 53 (81%) reported clinical recovery and 12 (19%) clinical failure.

In the group of patients treated with cefiderocol alone (*n* = 94), the clinical outcome was favorable in 65 patients (69%); in Bassetti et al., the outcome was undefined in 7 cases. In the combo-therapy group (*n* = 27), clinical success was achieved in 17 cases (63%).

Microbiological cure was of 37.8% with negative blood cultures and infection-free status in 62 cases. No breakthrough infections and 20.7% of recurrence were reported.

A total of 48 deaths were reported, but it is difficult to define infection versus non-infection-related deaths. In Bassetti et al., 34 of 101 patients receiving CFD died; the majority of the deaths occurred by the end of the study, so other causes other than treatment failure may have contributed. In Bleibtreu et al., two patients died due to the course of infection. In Falcone et al., a patient with COVID-19 infection died, while two burn patients died after more than 30 days from the beginning of CFD therapy. In Grande Perez et al., the patient died on day 230 due to an XDR P. aeruginosa-associated pneumonia. In Meschiari et al., two deaths were associated with both clinical and microbiological failures. For patients who did not survive in Rando et al., the main causes of death were respiratory failure (*n* = 3) and septic shock (*n* = 3).

## 4. Discussion

During the last decade, there has been an alarming global increase in the incidence and prevalence of carbapenem-resistant GN bacteria. The three most frequent carbapenem-resistant pathogens in this review were *Acinetobacter baumannii*, *Pseudomonas aeruginosa*, and *Klebsiella pneumoniae*. This is in accord with the European Centre for Disease Prevention and Control (ECDC) epidemiological report for 2020 [43]. Carbapenem resistance was common in *Acinetobacter* species (38%), *Pseudomonas aeruginosa* (17.8%), and *Klebsiella pneumoniae* (10%). In 2019, the U.S. Centers for Disease Control and Prevention (CDC) reported 13,100 cases of carbapenem-resistant *Enterobacteriaceae* and 8500 cases of carbapenem-resistant *Acinetobacter*; multidrug-resistant *Pseudomonas aeruginosa* were isolated in 32,600 cases [44].

According to our data, the most effective carbapenemases, in terms of carbapenem hydrolysis and geographical spread, are OXA, NDM, VIM, and KPC [8]. Among carbapenem-resistant OXA-type β-lactamase, OXA-23 and OXA-48 were the most recurrent. The most dominant groups of Metallo-β-Lactamases (MBLs) were NDM and VIM. Cases of KPC have also been reported, frequently associated with *Klebsiella Pneumoniae*.

The risk factors for the development of resistance to carbapenems are the same as reported in the literature [45]; moreover, at present, with the ongoing COVID19 pandemic, the presence of multi-resistant bacteria is widespread among patients admitted to intensive care with SARS-CoV-2 infection [46].

In this review, we discuss the cases of patients who developed several types of infections by carbapenem-resistant GN bacteria, and failed several antibiotic therapy lines. Finally, 70% of them recovered after compassionate treatment with CFD.

CFD is authorized for the treatment of the complicated urinary tract infections (cUTI), and hospital-acquired bacterial pneumonia and ventilator-associated bacterial pneumonia (HABP/VABP) caused by susceptible GN microorganisms [20]. The studies we have included report cases due to other conditions, such as osteo-articular infections, bacteremia or sepsis, intra-abdominal infections, skin infection, and endocarditis. During the clinical course, treatment options were extremely limited, and phenotypic testing was carried out on several antibiotics to explore any alternative treatment options. In the included studies, CFD MICs of the tested strains were inferior to the susceptibility breakpoint of ≤4 mg/L accepted for *Enterobacteriaceae*, *Pseudomonas* and *Acinetobacter*, and proposed by the Clinical & Laboratory Standards Institute (CLSI) (see Table 5) [23].

As reported in this review, the proposed dosing regimen for CFD is 2 g intravenously every 8 h and dosage adjustments for altered renal function are required [20]. For CFD-treated patients with a creatinine clearance of more than 120 mL/min, a regimen of 2 g every 6 h is used; in case of renal impairment or dialysis, the dosing regimen for CFD is reduced [20]. The proposed treatment duration is 7–14 days [20]. In the included studies, the duration of therapy was further prolonged up to 102 days in Dragher M et al. The mean duration of therapy was approximately 26 days and was guided by the patient’s clinical status.

The role of combination therapy for the treatment of severe multi-resistant GN infections has long been debated. In 31.8% of total cases, CFD was associated with different drugs, particularly with colistin. Colistin has been long considered the first-line therapy against MDR GN bacteria; however, alternative antimicrobials or combination regimens have been investigated in order to increase success rates due to the unpredictable pharmacodynamic/pharmacokinetic properties and the considerable kidney toxicity [47]. Additionally, in recent years colistin resistance significantly increased, causing a reduction in possible treatment options for multi-resistant GN infections.

Resistance to CFD was reported in six patients, all regarding *Pseudomonas aeruginosa*. Mechanisms through which resistance to CFD develops remain unclear. No direct correlation was observed between resistance to CFD and acquired carbapenemases. CFD is transported into the periplasmic space through siderophore iron transporters known as TonB-dependent receptors; mutations in the genes encoding for these transporters can lead to a loss of function for the receptors that are required for CFD import [48]. Associations between elevated cefiderocol MIC and β-lactamases have also been reported. Β-lactamases associated with CFD resistance are NDM and *Pseudomonas*-extended-resistant (PER) β-lactamases [49]. Mutations in the chromosomal ampC β-lactamase also are responsible for the development of resistance to CFD [50]. Among risk factors for resistance to CFD, empiric CFD use is discouraged unless there is a known risk factor for infection due to the presence of extensively drug-resistant pathogens. As with all other antimicrobials, failure to exercise careful antimicrobial stewardship may also compromise the long-term efficacy of CFD, as in the case of excessive use of ceftazidime/avibactam responsible to select for metallo-β-lactamases and for NDM [51]. Finally, the patient’s travel history and geographical location may represent risk factors for resistance to CFD.

CFD was generally well tolerated, with fewer side effects than existing alternative treatments for carbapenem-resistant bacteria. Among the patients under consideration, a total of 97 adverse reactions were observed. Only in three cases, CFD administration was discontinued, although a long treatment duration was prescribed. In cefiderocol trials, some data about adverse drug reactions are available. The most frequently reported AEs are elevated liver tests and hypokalemia, all reported in more than 10% of subjects receiving CFD [52]. Diarrhea, hypomagnesemia, atrial fibrillation, thrombocytopenia, and many others have been reported with an incidence ranging from 1% to 10% [52]. Severe or serious AEs are rare, corresponding to 2% and 4.7%, respectively. An increase in all-cause mortality is observed in CFD-treated patients compared to those treated with the best available therapy [20]. Concerning mortality, a total of 48 deaths were reported.

In conclusion, the success rate of CFD therapy was satisfactory. Almost 70% of patients showed clinical recovery, and of these nearly half showed negative blood cultures and infection-free status.

**Limitations**. This review presents some limitations. First, it was based on observational studies, the majority of which were case series or case reports. They are often excluded from systematic reviews due to the greater potential for bias. In this report, case series and observational studies contribute to the available evidence base, and their results supplement the limited evidence available from other studies. Second, a meta-analysis was not performed, due to the design of most of the studies (case report, case series) and the lack of a comparator.

## 5. Conclusions

Despite recent advances in the development of antibacterial agents, there is still an unmet need for antibacterial agents with an acceptable safety profile that are active against carbapenem-resistant GN organisms, especially against organisms producing carbapenemases. This review indicates that CFD is active against important GN organisms including *Enterobacteriaceae, P. aeruginosa*, and *A. baumannii*. CFD seems, moreover, to have a safe profile. Therefore, CFD could provide a useful alternative for the treatment of most infections due to carbapenem-resistant GN bacteria.

## Figures and Tables

**Figure 1 antibiotics-11-00904-f001:**
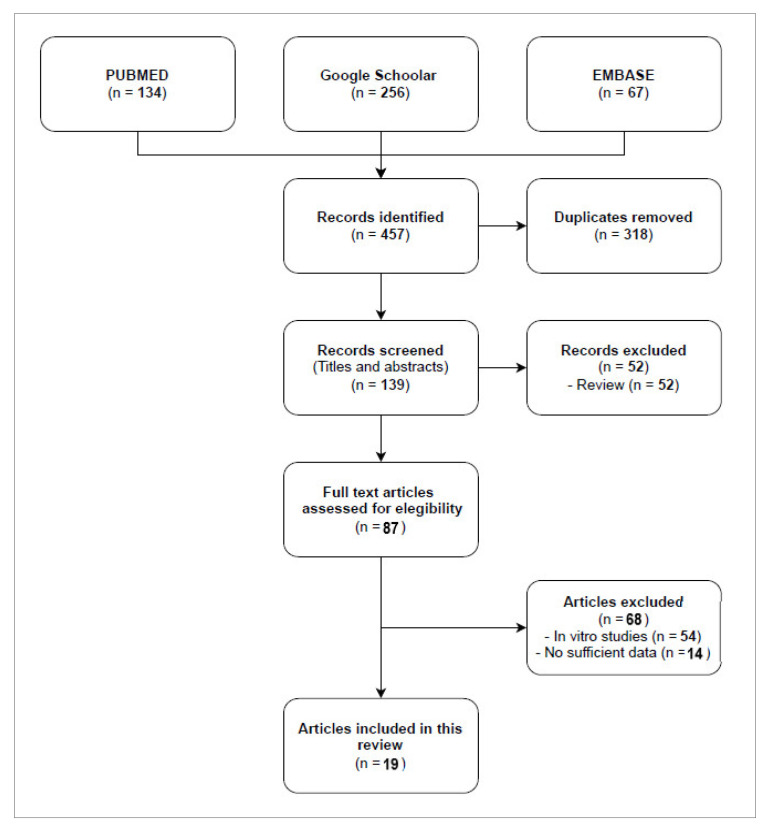
Flow diagram study selection process.

**Figure 2 antibiotics-11-00904-f002:**
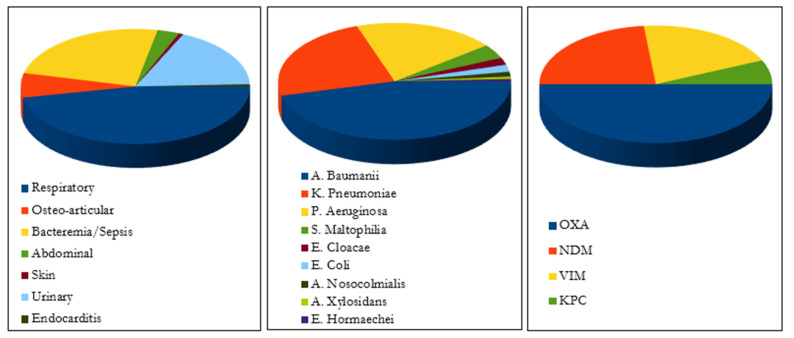
Type of infection, bacterial agents and carbapenemase enzymes.

**Figure 3 antibiotics-11-00904-f003:**
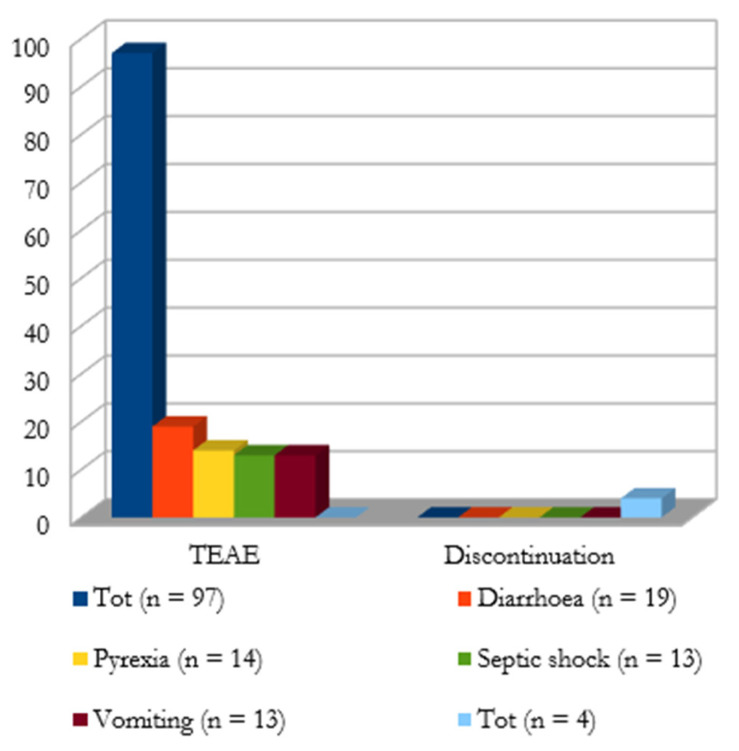
Treatment-emergent adverse events (TEAE) and discontinuation cases; clinical (external ring) and microbiological (internal ring) outcome.

**Table 1 antibiotics-11-00904-t001:** Breakpoint values of minimum inhibitory concentration (mg/L) for carbapenems according to guidelines in Europe (EUCAST) and the United States (CLSI).

	EUCAST						CLSI					
	*Enterobacteriaceae*		*Acinetobacter*		*Pseudomonas*		*Enterobacteriaceae*		*Acinetobacter*		*Pseudomonas*	
**Carbapenem**	**S**	**R**	**S**	**R**	**S**	**R**	**S**	**R**	**S**	**R**	**S**	**R**
**Doripenem**	≤1	≥4	≤1	≥2	≤1	≥2	≤1	≥4	≤2	≥8	≤1	≥8
**Ertapenem**	≤0.5	≥1	-	-	-	-	≤0.5	≥2	-	-	-	-
**Imipenem**	≤2	≥8	≤2	≥8	≤4	≥8	≤1	≥4	≤2	≥8	≤2	≥8
**Meropenem**	≤2	≥8	≤2	≥8	≤2	≥8	≤1	≥4	≤2	≥8	≤2	≥8

S, susceptible; R, resistant.

**Table 2 antibiotics-11-00904-t002:** Case reports of CR bacteria infection.

Author, Year	Study	No.	Country	Bacterial Agent	Carbapenemase	Site of Infection
**Alamarat ZI et al., 2020** [31]	CR	1	Nigeria	-Pseudomonas aeruginosa (*n* = 1)	-NDM-1 (*n* = 1)	-Osteomyelitis (*n* = 1)
**Bassetti et al., 2020** [24]	RCT	101 (80)	16 countries in North America, South America, Europe and Asia	-Acinetobacter baumannii (*n* = 37) -Klebsiella Pneumoniae (*n* = 27) -Pseudomonas aeruginosa (*n* = 12) -Stenotrophomonas maltophilia (*n* = 5) -Acinetobacter nosocomialis (*n* = 2) -Enterobacter cloacae (*n* = 2) -Eschierichia Coli (*n* = 2)	N/D	-Nosocomial pneumonia (*n* = 45) -BSI or sepsis (*n* = 30) -cUTI (*n* = 26)
**Bavaro DF et al., 2020** [32]	CR	1	Italy	-Pseudomonas aeruginosa (*n* = 1)	N/D	-Osteomyelitis (*n* = 1)
**Bleibtreu et al., 2021** [25]	OS	12	France	-Pseudomonas aeruginosa (*n* = 9) -Acinetobacter baumannii (*n* = 2) -Klebsiella pneumoniae (*n* = 1) -Enterobacter hormaechei (*n* = 1)	-VIM-2 (*n* = 3) -VIM-4 (*n* = 1) -OXA-23 (*n* = 2) -OXA-48 (*n* = 1) -OXA-836 (*n* = 1) -NDM-1 (*n* = 1)	-Respiratory tract (*n* = 10) -Intra-abdominal (*n* = 2), -Osteo-articular (*n* = 2), -Skin-and-skin structure (*n* = 1), -Urinary tract (*n* = 1).
**Bodro et al., 2021** [27]	CS	2	Spain	-Acinetobacter xylosidans (or xylosoxidans?) (*n* = 1) -Pseudomonas aeruginosa (*n* = 1)	N/D	-Bacteremia (*n* = 2)
**Carney et al., 2021** [33]	CR	1	USA	-Eschierichia Coli (*n* = 1)	-NDM-5 (*n* = 1)	-Osteomyelitis (*n* = 1)
**Cipko K et al., 2021** [34]	CR	1	Australia	-Acinetobacter baumannii (*n* = 1)	-OXA-23 (*n* = 1)	-Osteomyelitis (*n* = 1)
**Contreras DA et al., 2020** [35]	CR	1	USA	-Klebsiella pneumoniae (*n* = 1)	-NDM-1 (*n* = 1) -OXA-48 (*n* = 1)	-Abdominal infection (*n* = 1)
**Dragher M et al., 2020** [36]	CR	1	USA	-Acinetobacter baumannii (*n* = 1)	-OXA-23 (*n* = 1)	-Osteomyelitis (*n* = 1)
**Edgeworth JD et al., 2019** [37]	CR	1	Kuwait	-Acinetobacter baumannii (*n* = 1) -Klebsiella pneumoniae (*n* = 1)	-OXA-23 (*n* = 1) -OXA-48 (*n* = 1) -OXA-51 (*n* = 1)	-Endocarditis (*n* = 1)
**Falcone et al., 2020** [28]	CS	10	Italy	-Acinetobacter baumannii (*n* = 8) -Klebsiella pneumoniae (*n* = 3) -Stenotrophomonas maltophilia (*n* = 1)	-NDM (*n* = 3)	-VAP (*n* = 4) -BSI (*n* = 6)
**Grande Perez C et al., 2021** [38]	CR	1	Belgium	-Pseudomonas aeruginosa (*n* = 1)	-VIM (*n* = 1)	-Pancreatitis (*n* = 1)
**Meschiari M et al., 2021** [29]	CS	17	Italy	-Pseudomonas aeruginosa (*n* = 17)	N/D	-VAP (*n* = 7) -HAP (*n* = 1) -Peritonitis (*n* = 3) -Cholangitis (*n* = 1) -Osteomyelitis (*n* = 1) -Meningitis (*n* = 1) -Skin infection (*n* = 1) -Empyema (*n* = 1) -Primary bacteremia (*n* = 1)
**Oliva A et al., 2020** [26]	CS	3	Italy	-Acinetobacter baumannii (*n* = 3) -Klebsiella pneumoniae (*n* = 1) -Pseudomonas aeruginosa (*n* = 1)	N/D	-VAP (*n* = 1) -BSI (*n* = 1) -Spondylodiscitis (*n* = 1)
**Rando E et al., 2022** [39]	OS	13	Italy	-Acinetobacter baumannii (*n* = 13) -Klebsiella pneumoniae (*n* = 2) -Pseudomonas aeruginosa (*n* = 2)	N/D	-VAP (*n* = 10) -HAP (*n* = 3)
**Simeon S et al., 2020** [40]	CR	1	France	-Enterobacter hormaechei (*n* = 1)	N/D	-knee prosthetic joint infection (*n* = 1)
**Stevens WS et al., 2019** [41]	CR	1	USA	-Pseudomonas aeruginosa (*n* = 1)	N/D	-Abdominal infection (*n* = 1)
**Trecarichi EM et al., 2019** [42]	CR	1	Italy	-Acinetobacter baumannii (*n* = 1) -Klebsiella pneumoniae (*n* = 1)	KPC	-VAP (*n* = 1) -BSI (*n* = 1)
**Zingg S et al., 2020** [30]	CS	3	Swisse	-A. baumannii (*n* = 3) -E. cloacae KPC (*n* = 1) -Pseudomonas aeruginosa (*n* = 1)	-OXA-23 (*n* = 2) -OXA-40 (*n* = 1) -OXA-58 (*n* = 1) -NDM (*n* = 1) -VIM (*n* = 1) -KPC	-Acute osteomyelitis (*n* = 1) -Postoperative implant-associated surgical site infection (*n* = 1) -Pleural empyema (*n* = 1)

**Table 3 antibiotics-11-00904-t003:** Baseline characteristics of patients.

	No.
**Patients**	160
**M/F**	111/49
**Age** ± SD (years)	58.8 ± 15.8
**Race** -White -Asian -Black or African American -Other -*n*/D	97 31 1 11 32
**Comorbidities** -Renal disease (renal transplant) -Diabetes -Chronic pulmonary disease -Cancer -Congestive/ischemic heart disease -Arterial hypertension -Vascular disease -Hepatitis -Cerebral or neurological disease -Blood disease -Atrial fibrillation -Pancreatitis -Endocarditis -Hypothyroidism -Gout arthritis -Recurrent infections in hip replacement	44 (3) 44 43 31 17 16 13 12 6 3 2 1 1 1 1 1
**Country of infection** -Afghanistan -Australia -Belgium -Columbia -France -Italy -Kuwait -Nigeria -Serbia -Spain -Thailand -US	1 1 1 1 2 6 1 1 1 1 1 3

**Table 4 antibiotics-11-00904-t004:** Susceptibility Patterns (minimal inhibitory concentration (MIC) or zone of inhibition (ZOI)) of Treated Gram-Negative Bacteria.

Study	Bacterial Agent	AMK	AZT	CEF	CFD	CZA	CIP	COL	FOM	GEN	IPM	MEM	TZP	TGC	TOB
Alamarat ZI et al.	*P. aeruginosa*	<32	6	>16	4	>256	>2	0.75	NA	>8	R	>8	>64	>4	>8
Bassetti et al.	*A. baumannii*	NA	NA	NA	1	NA	NA	NA	NA	NA	R	R	NA	NA	NA
	*K. pneumoniae*	NA	NA	NA	4	NA	NA	NA	NA	NA	R	R	NA	NA	NA
	*P. aeruginosa*	NA	NA	NA	2	NA	NA	NA	NA	NA	R	R	NA	NA	NA
Bavaro DF et al.	*P. aeruginosa*	NA	NA	16 (R)	0.5 (S) 27 mm	>8 (R)	>2 (R)	1 (S)	32 (S)	>8 (R)	>8 (R)	>8 (R)	>64 (R)	NA	>8 (R)
Bleibtreu et al.	*P. aeruginosa*	64 (R)	R	R	2 (S)	32 (R)	32 (R)	2 (S)	NA	>256 (R)	32 (R)	32 (R)	R	16 (R)	>256 (R)
	*A. baumannii*	16 (S)	R	R	1 (S)	32 (R)	32 (R)	2 (S)	NA	>256 (R)	32 (R)	32 (R)	R	2	3 (S)
	*A. baumannii*	>256 (R)	R	R	0.5 (S)	32 (R)	32 (R)	1 (S)	NA	>256 (R)	32 (R)	32 (R)	R	4	>256 (R)
	*P. aeruginosa*	>256 (R)	R	R	4 (S)	32 (R)	32 (R)	4 (R)	NA	>256 (R)	2	16 (R)	R	8 (R)	>256 (R)
	*P. aeruginosa*	>256 (R)	R	R	2 (S)	32 (R)	32 (R)	2 (S)	NA	>256 (R)	32 (R)	32 (R)	R	16 (R)	>256 (R)
	*E. hormaechei*	16 (R)	R	R	1 (S)	8 (S)	32 (R)	0.5 (S)	NA	>256 (R)	8 (R)	16 (R)	R	1 (S)	48 (R)
	*K. pneumoniae*	4 (R)	R	R	0.5 (S)	≤0.25 (S)	1.5 (R)	1 (S)	NA	0.5 (S)	2 (I)	2 (R)	R	2	6 (S)
	*P. aeruginosa*	>256 (R)	R	R	4 (S)	32 (R)	32 (R)	2 (S)	NA	>256 (R)	32 (R)	32 (R)	R	8 (R)	>256 (R)
	*P. aeruginosa*	>256 (R)	R	R	8 (I)	32 (R)	32 (R)	2 (S)	NA	16 (R)	32 (R)	32 (R)	R	8 (R)	32 (R)
	*P. aeruginosa*	>256 (R)	R	R	16 (R)	32 (R)	32 (R)	64 (R)	NA	8 (I)	32 (R)	16 (R)	R	8 (R)	>256 (R)
	*P. aeruginosa*	>256 (R)	R	R	16 (R)	32 (R)	32 (R)	2 (S)	NA	>256 (R)	2	16 (R)	R	8 (R)	>256 (R)
	*P. aeruginosa*	>256 (R)	R	R	>32 (R)	32 (R)	32 (R)	2 (S)	NA	>256 (R)	32 (R)	32 (R)	R	8 (R)	>256 (R)
	*P. aeruginosa*	16 (S)	R	R	16 (R)	32 (R)	4 (R)	2 (S)	NA	3 (S)	32 (R)	32 (R)	R	16 (R)	1 (S)
Bodro et al.	*A. xylosidans*	NA	NA	NA	21 mm (S)	NA	>2 (R)	1 (S)	NA	NA	R	>16 (R)	2/4 (S)	2 (S)	NA
	*P. aeruginosa*	NA	NA	NA	23 mm (S)	>8/4 (R)	>1 (R)	2 (S)	NA	NA	R	>8 (R)	32/4 (R)	NA	>4 (R)
Carney et al.	*E. coli*	8	>32	>16	2 (S)	>64	>4	4	NA	NA	16	64	NA	0.5	NA
Cipko K et al.	*A. baumannii*	≤16 (S)	>84	>2 (R)	0.5	25 (R)	>1 (R)	1 (S)	>32 (R)	>2 (R)	>32 (R)	>32 (R)	>16/2 (R)	4 (R)	NA
Contreras DA et al.	*K. pneumoniae*	>32 (R)	>32 (R)	>32 (R)	21 mm (S)	>32 (R)	≥2 (R)	≤2 (R)		>16 (R)	>16 (R)	>16 (R)	128 (R)	1 (S)	>16 (R)
Dragher M et al.,	*A. baumannii*	>32 (R)	>16 (R)	>16 (R)	23 mm (S)	15 mm (R)	>2 (R)	≤2 (S)	17 mm (S)	>8 (R)	6 mm (R)	>8 (R)	NA	6 mm (R)	>8 (R)
Edgeworth JD et al.	*P. aeruginosa*	S	NA	NA	21.3 mm (S)	R	NA	S	NA	S	R	>32 (R)	NA	NA	NA
Falcone et al.	*A. baumannii*	NA	NA	NA	0.25	NA	NA	NA	NA	NA	R	R	NA	NA	NA
	*A. baumannii*	NA	NA	NA	0.5	NA	NA	NA	NA	NA	R	R	NA	NA	NA
	*A. baumannii*	NA	NA	NA	0.5	NA	NA	NA	NA	NA	R	R	NA	NA	NA
	*A. baumannii*	NA	NA	NA	0.5	NA	NA	NA	NA	NA	R	R	NA	NA	NA
	*A. baumannii*	NA	NA	NA	0.25	NA	NA	NA	NA	NA	R	R	NA	NA	NA
	*A. baumannii*	NA	NA	NA	0.5	NA	NA	NA	NA	NA	R	R	NA	NA	NA
	*S. maltophilia* + *K. pneumoniae*	NA	NA	NA	0.5/1	NA	NA	NA	NA	NA	R	R	NA	NA	NA
	*K. pneumoniae*	NA	NA	NA	1	NA	NA	NA	NA	NA	R	R	NA	NA	NA
	*A. baumannii* + *K. pneumoniae*	NA	NA	NA	0.12/2	NA	NA	NA	NA	NA	R	R	NA	NA	NA
	*A. baumannii*	NA	NA	NA	0.5	NA	NA	NA	NA	NA	R	R	NA	NA	NA
Grande Perez C et al.	*P. aeruginosa*	NA	NA	NA	8 (R)	NA	NA	S	NA	NA	R	R	NA	NA	NA
Meschiari M et al.	*P. aeruginosa*	4	NA	16	≤2	16	0.5	NA	NA	2	>8	>8	16	NA	NA
	*P. aeruginosa*	≤32	NA	32	1	32	1	NA	32	2	>8	32	64	NA	NA
	*P. aeruginosa*	32	NA	16	0.25	16	0.5	NA	>64	≤1	8	≥16	32	NA	NA
	*P. aeruginosa*	4	NA	16	0.5	16	1	NA	>64	NA	> 8	16	≥128	NA	NA
	*P. aeruginosa*	4	NA	8	≤2	2	2	NA	NA	2	2	2	32	NA	NA
	*P. aeruginosa*	2	NA	16	≤2	16	0.12	NA	NA	≤1	>8	>8	>64	NA	NA
	*P. aeruginosa*	≤1	NA	NA	≤2	32	>2	NA	Na	≤1	>8	>8	>64	NA	NA
	*P. aeruginosa*	2	NA	16	NA	16	1	NA	128	≤1	>8	32	32	NA	NA
	*P. aeruginosa*	8	NA	>16	≤2	>32	> 2	NA	64	>8	>8	> 8	>64	NA	NA
	*P. aeruginosa*	2	NA	>32	0.12	>32	0.5	NA	32	2	>8	64	16	NA	NA
	*P. aeruginosa*	2	NA	>32	0.5	≥64	0.25	NA	64	≤1	NA	32	≥128	NA	NA
	*P. aeruginosa*	8	NA	≥32	NA	≥64	1	NA	>256	4	>8	64	≥128	NA	NA
	*P. aeruginosa*	8	NA	≥32	1	≥64	≥4	NA	>64	≥16	NA	16	32	NA	NA
	*P. aeruginosa*	4	NA	NA	≤2	16	>2	NA	128	4	NA	>8	32	NA	NA
	*P. aeruginosa*	4	NA	16	≤2	>32	0.5	NA	NA	2	>8	>8	>64	NA	NA
	*P. aeruginosa*	4	NA	16	≤2	8	>2	NA	64	>8	>8	>8	>64	NA	NA
	*P. aeruginosa*	4	NA	16	≤2	>32	0.12	NA	NA	4	>8	>8	>64	NA	NA
Oliva A et al.	*A. baumannii*	NA	NA	NA	S	NA	NA	2	NA	NA	R	R	NA	4	NA
	*A. baumannii*	NA	NA	NA	S	NA	NA	0.5 (S)	NA	NA	R	R	NA	4	NA
	*A. baumannii*	NA	NA	NA	S	NA	NA	0.5 (S)	NA	NA	R	R	NA	2	NA
Simeon S et al.	*E. hormaechei*	16 (R)	>32 (R)	>32 (R)	1 (S)	8 (S)	>32 (R)	0.5 (S)		>256 (R)	8 (R)	16 (R)	128 (R)	1 (S)	48 (R)
Rando E et al.	*A. baumannii* + *P. aeruginosa*	NA	NA	NA	S	NA	NA	S	NA	NA	R	R	NA	NA	NA
	*A. baumannii*	NA	NA	NA	S	NA	NA	S	NA	NA	R	R	NA	NA	NA
	*A. baumannii* + *P. aeruginosa*	NA	NA	NA	S	NA	NA	S	NA	NA	R	R	NA	NA	NA
	*A. baumannii*	NA	NA	NA	S	NA	NA	S	NA	NA	R	R	NA	NA	NA
	*A. baumannii*	NA	NA	NA	S	NA	NA	S	NA	NA	R	R	NA	NA	NA
	*A. baumannii*	NA	NA	NA	S	NA	NA	S	NA	NA	R	R	NA	NA	NA
	*A. baumannii*	NA	NA	NA	S	NA	NA	S	NA	NA	R	R	NA	NA	NA
	*A. baumannii*	NA	NA	NA	S	NA	NA	S	NA	NA	R	R	NA	NA	NA
	*A. baumannii*	NA	NA	NA	S	NA	NA	S	NA	NA	R	R	NA	NA	NA
	*A. baumannii*	NA	NA	NA	S	NA	NA	S	NA	NA	R	R	NA	NA	NA
	*A. baumannii* + *K. pneumoniae*	NA	NA	NA	S	NA	NA	S	NA	NA	R	R	NA	NA	NA
	*A. baumannii* + *K. pneumoniae*	NA	NA	NA	S	NA	NA	S	NA	NA	R	R	NA	NA	NA
	*A. baumannii*	NA	NA	NA	S	NA	NA	S	NA	NA	R	R	NA	NA	NA
Stevens WS et al.	*P. aeruginosa*	8 (S)	>32 (R)	>16 (R)	0.12 (S)	32 (R)	>4 (R)	1 (S)	NA	>16 (S)	32 (R)	32 (R)	8 mm (R)	NA	>16 (R)
Trecarichi EM et al.	*A. baumannii*	NA	NA	NA	S	NA	NA	0.5 (S)	NA	NA	R	R	NA	NA	NA
	*K. pneumoniae*	NA	NA	NA	S	4 (S)	NA	0.5 (S)	NA	2 (S)	R	R	NA	NA	NA
Zingg S et al.	*A. baumannii*	R	NA	R	23 mm (S)	R	R	S	S	NA	R	R	NA	S	R
	*E. cloacae*	R	R	I	14 mm (R)	S	R	S	R	NA	I	R	NA	S	S
	*P. aeruginosa*	R	R	R	24 mm (S)	R	R	S	S	NA	R	R	NA	NA	R
	*A. baumanni*	R	R	R	18 mm (S)	R	R	S	R	NA	R	R	NA	R	R
	*A. baumanni*	I	R	R	20 mm (S)	R	R	S	R	NA	R	R	NA	R	R

AMK, amikacin; AZT, aztreonam; CEF, cefepime; CFD, cefiderocol; CZA, cefazolin; CIP, ciprofloxacin; COL, colistin; FOM, fosfomycin; GEN, gentamicin; IPM, imipenem; MEM, meropenem; TZP, piperacillin/tazobactam; TGC, tigecycline; TOB, tobramycin; S, sensible; R, resistant; I, intermediate; NA, not available.

**Table 5 antibiotics-11-00904-t005:** Breakpoint values of minimum inhibitory concentration (mg/L) for cefiderocol according to guidelines in Europe (EUCAST) and the United States (CLSI).

	EUCAST									CLSI								
	*Enterobacteriaceae*			*Acinetobacter*			*Pseudomonas*			*Enterobacteriaceae*			*Acinetobacter*			*Pseudomonas*		
	S	I	R	S	I	R	S	I	R	S	I	R	S	I	R	S	I	R
**CFD**	<2	–	>2	–	–	–	<2	–	>2	<4	8	>16	<4	8	>16	<4	8	>16

## Data Availability

Not applicable.

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
