# Peer review of "Cefiderocol for Carbapenem-Resistant Bacteria: Handle with Care! A Review of the Real-World Evidence"

_antibiotics, 2022, doi:10.3390/antibiotics11070904_

Round 1

Reviewer 1 Report

The manuscript is a review article investigating the use of cefiderocol in the treatment of infections produced by carbapenem-resistant bacteria. Unfortunately, it has too many flaws and needs a lot of improvement.

The manuscript seems to be a draft and not ready for submission. Colored or bold sections can be seen all over the text. There is also a discrepancy between the text and the abstract concerning the number of articles included in the review. The four case series are five. The sentence “the majority of them recovered” (line 307) doesn’t match with “the success rate of CFD therapy was satisfactory” (line 371).

The most frequently used cefiderocol regimen was following the EUCAST indications; still, studies that did not reveal the cefiderocol regimen were also included, even if they could bring diversity and make comparison very difficult. Patients receiving combination therapy were also included, even if the exact effects of cefiderocol can’t be assessed. Stricter inclusion criteria should be used in order to demonstrate correlations with the recommended cefiderocol regimen. What is the impact of the differences between EUCAST and CLSI in the assessment of the strains’ susceptibility to cefiderocol and on the patients’ evolution?

There are inaccurate phrases like, for example, “Carbapenems inhibit β-lactamase enzyme penicillinase” (line 47) or the suggestion that carbapenem-resistant bacteria are only due to carbapenemase production (lines 50-52) or the imipenem nonsusceptibility of Morganellaceae (lines 140-142).

Tables and figures are not explicit, sometimes illegible.

Table 4 is confusing; it mixes MIC values with ZOI values and should be modified. Why there is not always an interpretation of the MIC/ZOI detected?

Figure 2 should be redone.

Figure 3 is impossible to read. Are TEAE strictly related to cefiderocol or other antibiotics/medicines have also been used in these patients?

The names of bacteria are often incorrectly written.

English language needs a lot of improvement culminating with “curse” instead of “course” (line 267).

Author Response

- Colored or bold sections can be seen all over the text: MODIFIED

- Discrepancy between the text and the abstract concerning the number of articles included in the review: CORRECETD

- The four case series are five: CORRECTED

- The sentence “the majority of them recovered” (line 307) doesn’t match with “the success rate of CFD therapy was satisfactory” (line 371): MODIFIED

- The most frequently used cefiderocol regimen was following the EUCAST indications; still, studies that did not reveal the cefiderocol regimen were also included, even if they could bring diversity and make comparison very difficult. Patients receiving combination therapy were also included, even if the exact effects of cefiderocol can’t be assessed. Stricter inclusion criteria should be used in order to demonstrate correlations with the recommended cefiderocol regimen: THE NUMBER OF CASES IS LIMITED (172), SO WE DECIDED TO INCLUDED ALL THE CASES IN WHICH CEFIDEROCOL WAS USED, ALSO IN COMBINATION THERAPY.

- What is the impact of the differences between EUCAST and CLSI in the assessment of the strains’ susceptibility to cefiderocol and on the patients’ evolution? AS SHOWED IN OTHER STUDIES COMPARING THE TWO SYSTEMS, THE RESULTS SHOW COMPARABLE ANTIBIOTIC SUSCEPTIBILITY PATTERNS BETWEEN CLSI AND EUCAST BREAKPOINTS. GIVEN THAT EUCAST GUIDELINES ARE FREELY AVAILABLE, IT MAKES IT EASIER FOR LABORATORIES TO HAVE AN UPDATED AND READILY AVAILABLE REFERENCE FOR INTERPRETING ANTIBIOTIC SUSCEPTIBILITIES.

- There are inaccurate phrases like, for example, “Carbapenems inhibit β-lactamase enzyme penicillinase” (line 47) or the suggestion that carbapenem-resistant bacteria are only due to carbapenemase production (lines 50-52) or the imipenem nonsusceptibility of Morganellaceae (lines 140-142): MODIFIED

- Tables and figures are not explicit, sometimes illegible: MODIFIED

- Table 4 is confusing; it mixes MIC values with ZOI values and should be modified. Why there is not always an interpretation of the MIC/ZOI detected? NOT ALWAYS AVAILABLE (MIC OR ZOI) IN STUDIES INCLUDED

- Figure 2 should be redone: MODIFIED

- Figure 3 is impossible to read. Are TEAE strictly related to cefiderocol or other antibiotics/medicines have also been used in these patients? MODIFIED. TEAE ARE RELATED TO CFD

- The names of bacteria are often incorrectly written: MODIFIED

- English language needs a lot of improvement culminating with “curse” instead of “course” (line 267): MODIFIED (IN BLUE)

Reviewer 2 Report

I do not find this manuscript suitable for publication in this form. The English language needs great improvements, the font and style throughout the manuscript need to be changed. The literature review should be more extensive and references up to date. Furthermore, the tables and figures are of low quality.

Therefore, I could not judge the quality of the study itself before this manuscript is improved.

Author Response

Dear Reviewer:

- English language was improved;

- the font and style throughout the manuscript was changed.

- The last article was “Meschiari M, Volpi S, Faltoni M, et al. Real-life experience with compassionate use of cefiderocol for difficult-to-treat resistant Pseudomonas aeruginosa (DTR-P) infections. JAC Antimicrob Resist. 2021;3(4):dlab188. .

- Tables and figures were modified.

Round 2

Reviewer 2 Report

The manuscript is appropriate for publication 

Author Response

Dear Reviewer:

- Line 16 e to carpapenemase? → CORRECTED
- Line 25: success rate of CFD therapy was satisfactory and almost 70% of patients showed clinical
recovery. ? clinical cure was assess in almost 70% of patients and overall-mortality was...? →
MODIFIED

- line 262: As already said, Please delete this sentence:" In our review, no death was considered to be related to CFD". → DELETED
- it's the most important e clinical point; the distinction from infection versus non-infection related deaths…a suggestion could be: concerning mortality ...A total of 48 deaths were reported. →
MODIFIED
– In general check for bacteria names because they are often incorrectly written. →
CHECKED
